# Generative Adversarial Neural Operators

## Abstract

We propose the generative adversarial neural operator (GANO), a generative model paradigm for learning probabilities on infinite-dimensional function spaces. The natural sciences and engineering are known to have many types of data that are sampled from infinite-dimensional function spaces, where classical finite-dimensional deep generative adversarial networks (GANs) may not be directly applicable. GANO generalizes the GAN framework and allows for the sampling of functions by learning push-forward operator maps in infinite-dimensional spaces. GANO consists of two main components, a generator neural operator and a discriminator neural functional. The inputs to the generator are samples of functions from a user-specified probability measure, e.g., Gaussian random field (GRF), and the generator outputs are synthetic data functions. The input to the discriminator is either a real or synthetic data function. In this work, we instantiate GANO using the Wasserstein criterion and show how the Wasserstein loss can be computed in infinite-dimensional spaces. We empirically study GANO in controlled cases where both input and output functions are samples from GRFs and compare its performance to the finite-dimensional counterpart GAN. We empirically study the efficacy of GANO on real-world function data of volcanic activities and show its superior performance over GAN. Furthermore, we find that for the function-based data considered, GANOs are more stable to train than GANs and require less hyperparameter optimization.

## 1 Introduction

Generative models are one of the most prominent paradigms in machine learning for analyzing unsupervised data. To date, there has been considerable success in developing deep generative models for finite-dimensional data (Goodfellow et al., 2014; Kingma & Welling, 2013; Dinh et al., 2014; Radford et al., 2015). Generative adversarial networks (GANs) are among the most successful generative models with rich theoretical and empirical developments (Arjovsky et al., 2017; Liu et al., 2017). The empirical success of GANs has been mainly within finite-dimensional data regimes; there has been relatively little progress on developing generative models for infinite-dimensional spaces–and importantly–function spaces. This is the case despite the fact that many fields of science and engineering, including seismology, computational fluid dynamics, aerodynamics, physics, and atmospheric sciences, work primarily with data that live in function spaces.

In this paper, we study the problem of generative models in function spaces. We propose generative adversarial neural operator (GANO), a deep learning-based approach that enables the learning of probabilities on function spaces, and allows for efficient sampling from such learned models. GANOs generalize the GAN paradigm to function spaces, and in particular, to separable Polish and Banach spaces. GANO, unlike traditional kernel density estimation methods, is computationally tractable, works on general spaces, and does not require the existence of a density nor the assumption of defined underlying measures for density (Rosenblatt, 1956; Parzen, 1962; Craswell, 1965)[1]. Another line of work proposes to use neural stochastic differential equation (SDE) solver (Tzen & Raginsky, 2019) to generate temporal signal function with finite dimensional co-domain (Kidger et al., 2021). However, while the generated signals are infinite dimensional objects, the lost construction in the mentioned work is still for finite dimensional spaces, for grid evaluation points, therefore, making the learned generative model implicitly yet for finite dimensional domains. Such generative

---

[1]In finite dimensional spaces, it is conventional and standard to define density with respect to Lebesgue measures. However, in the infinite dimensional cases considered in this paper, Lebesgue measures do not exists and a density, if exists, needs to be defined with respect to a user-defined measure that the users need to argue for its relevance.

| Models | **GANO** | **GAN** |
|---|---|---|
| Input/output spaces | Function Spaces | Euclidean spaces |
| Input measure | Gaussian Random Fields | Multivariate random variables |
| Controls | length scale, variance, energy, etc. | dimension, variance, etc. |

Table 1: GANOs and GANs

models require an underlying SDE solver to solve the temporal equation and is only designed for temporal data.

GANO consists of two main components, a generator neural operator and a discriminator neural functional. GANO architecture is empowered by neural operators, which are maps between function spaces (Li et al., 2020b). The generator neural operator receives a function sampled from a Gaussian random field (GRF) and outputs a function sample. This is in contrast to GAN, where the input is a sample from a finite-dimensional multivariate random variable and the output is a finite-dimensional object. The efficiency of traditional sampling methods from GRFs enables GANO to be considered as a computationally efficient generative model. The discriminator neural functional consists of a neural operator followed by an integral function. The discriminator receives either synthetic or real data as input and outputs a scalar. For the architecture choices in the generator, we use the efficient implementation of U-shaped neural operators (*U-NO*) (Rahman et al., 2022) and use Fourier integration layers, termed Fourier neural operator (FNO) (Li et al., 2020a) layers to construct push-forward maps from GRFs to the desired probability over function data. We use a similar architecture for the discriminator neural function and use a three-layered neural network to implement the integral functional layer.

The effective dimension of the output function space can be controlled by restricting the effective dimension of the GRF, e.g. by increasing the length scale of the defining covariance function. This is in contrast to GANs where the dimension of the input space controls the dimension of the output manifold. Table 1 compares the settings of GANOs and GANs. Since finite-dimensional spaces are special cases of infinite-dimensional spaces, and multi-variate Gaussian is a reduction of GRFs, then, GAN is a special case of GANO.

We construct a series of controlled empirical study to assess the performance of GANO. To maintain full control of the data characteristics and complexity of the task at hand, we generate the data itself using GRFs of varying complexities. We show that GANO can learn probability measures on function spaces. One important example is when the data is generated from a mixture of GRFs; GANO reliably recovers the measure, while GAN collapses to a mode. We show that as the roughness/noisiness of the input GRF is increased, GANO properly learns to generate functions from the underlying data probability, while if the input GRF generates smooth or nearly fixed-value functions, the trained models lose the ability to properly capture the data measure.

We extend our empirical study to satellite remote sensing observations of an active volcano, where each data point is the phase of a complex-valued function defined on a 2D domain (Rosen et al., 2012). This is a real world function dataset in which each data point represents $\sim$ millimeter-scale changes in the surface of a volcano at a spatial resolution of $\sim$ 70 meters, measured every 12 days. This dataset constitutes a noisy and challenging function dataset for GANO and GAN training. We show that GANO learns to generate functions on par with the real dataset while GAN fails in generating these volcanic phase functions.

We release the code to generate the data sets in the first part of the empirical study. For the purpose of bench-marking, we also release the processed volcano dataset, which is ready to be deployed in future studies. We also release the implementation code along with the training procedure.

## 2 Related Works

The original GAN formulation can be interpreted as an adversarial game procedure in which the Jensen–Shannon divergence between a synthetic distribution, implicitly defined by a generator model, and a real data distribution is minimized (Goodfellow et al., 2014). However, models trained with a Jensen-Shannon objective function require substantial tuning, suffer from stability issues, and are notoriously difficult

to scale (Radford et al., 2015). Considerable work has therefore been devoted to developing novel architectures, improving the formulation, and enhancing the theoretical understanding. In particular, the Wasserstein GAN (WGAN) allows for a more stable training scheme, is less sensitive to hyperparameter and architectural choices, and provides a loss function that correlates with output quality (Arjovsky et al., 2017). The WGAN formulation is often understood as an attempt to minimize the Wasserstein or Earth Mover's distance between the synthetic and real data distributions. In Adler & Lunz (2018), a rigorous theoretical extension of WGANs along with theoretically grounded choices of hyperparameters are presented, which the present paper follows.

To learn densities over function spaces, non-parametric density estimation with $\delta$-sequences on separable Banach spaces and topological groups has been studied (Rao, 2010; Craswell, 1965). Heuristic kernel density estimation for infinite-dimensional spaces was also developed (Dabo-Niang, 2004). Such methods assume the existence of a density with respect to (sometimes unspecified) base measures (Lebesgue measures are undefined for infinite-dimensional spaces) and impose strong assumptions on the metric and similarity of the output spaces. Moreover, learning the density does not provide matching algorithmic sampling methods from such infinite-dimensional spaces.

Pioneering work by (Li et al., 2020b) generalized the notion of neural networks to infinite-dimensional spaces and introduced the concept of neural operators, a novel composable architecture that is able to learn mappings between functions spaces. (Li et al., 2020a) showed that neural operators could be efficiently implemented as a series of convolutions performed in the Fourier domain of the input function. It has also been shown that any complex operator can be approximated by neural operators, which are compositions of linear integral operators and non-linear activation functions (Kovachki et al., 2021). Neural operators have been successfully used for learning the solution spaces of Partial Differential Equations (PDE). FNOs have been used to learn the solutions to the Accustic Wave-equation in two spatial dimensions (Yang et al., 2021). Operator learning has transformed the field of physics-informed machine learning. (Li et al., 2021; 2020a) and improvements in the underlying architecture have allowed neural operators to learn complex solutions to multiphase flow problems (Wen et al., 2021).

## 3 Generative Models in Function Spaces

One of the requirements to develop a stable model that maps an input probability measure to a general probability measure defined on infinite dimensional spaces is to have an infinite-dimensional input space. In this section, we describe the setting of such maps and propose GANO, a deep learning approach for learning generative models in infinite-dimensional function spaces. We propose GANO by extending the Wasserstein GAN formulation (Gulrajani et al., 2017) with a gradient penalty term applied to an infinite-dimensional setting.

### 3.1 GANO

Let $\mathcal{A}$ and $\mathcal{U}$ denote Polish function spaces, such that for any $a \in \mathcal{A}$, $a : D_{\mathcal{A}} \to \mathbb{R}^{d_{\mathcal{A}}}$, and for $u \in \mathcal{U}$, $u : D_{\mathcal{U}} \to \mathbb{R}^{d_{\mathcal{U}}}$. Let $\boldsymbol{G}$ denote a space of operators and for any operator $\mathcal{G} \in \boldsymbol{G}$, we have $\mathcal{G} : \mathcal{A} \to \mathcal{U}$, an operator map from $\mathcal{A}$ to $\mathcal{U}$. Let $\boldsymbol{L}$ denote a space of functionals such that for any functional $d \in \boldsymbol{L}$, we have $d : \mathcal{U} \to \mathbb{R}$, a functional map from $\mathcal{U}$ to $\mathbb{R}$.

Let $(\mathcal{A}, \sigma(\mathcal{A}), P_{\mathcal{A}})$ denote a probability space induced by a GRF on the function space $\mathcal{A}$, and $(\mathcal{U}, \sigma(\mathcal{U}), P_{\mathcal{U}})$ denote the probability space on the function spaces $\mathcal{U}$ that the real data is generated from. For a given function space $\mathcal{U}$, let $\mathcal{U}^*$ denote the dual space of $\mathcal{U}$. When $\mathcal{U}$ is also a Banach space, and $\mathcal{G}$ is Fréchet differentiable, we define $\partial \mathcal{G}$ as the Fréchet derivative of $\mathcal{G}$. For the measure $\mathbb{P}_{\mathcal{U}}$ and the pushforward measure of $\mathbb{P}_{\mathcal{A}}$ under map $\mathcal{G}$, i.e., $\mathcal{G}\sharp\mathbb{P}_{\mathcal{A}}$, we define the Wasserstein distance as follows,

$$W(\mathbb{P}_{\mathcal{U}}, \mathcal{G}\sharp\mathbb{P}_{\mathcal{A}}) = \sup_{d : d \in \boldsymbol{L}, Lip(d) \leq 1} \mathbb{E}_{\mathbb{P}_{\mathcal{U}}}[d] - \mathbb{E}_{\mathcal{G}\sharp\mathbb{P}_{\mathcal{A}}}[d] \tag{1}$$

For the dual space $\mathcal{U}^*$, we have that $Lip(d) \leq 1 \Leftrightarrow \|\partial d(u)\|_{\mathcal{U}^*} \leq 1, \ \forall u \in \mathcal{U}$. Therefore, we write the constraint in the form of an extra penalty part in the objective function, i.e.,

$$\inf_{\mathcal{G} \in \boldsymbol{G}} \sup_{d \in \boldsymbol{L}} \mathbb{E}_{\mathbb{P}_{\mathcal{U}}}[d] - \mathbb{E}_{\mathcal{G}\sharp\mathbb{P}_{\mathcal{A}}}[d] + \lambda \mathbb{E}_{\mathbb{P}'_{\mathcal{A}}}(\|\partial d\|_{\mathcal{U}^*} - 1)^2 \tag{2}$$

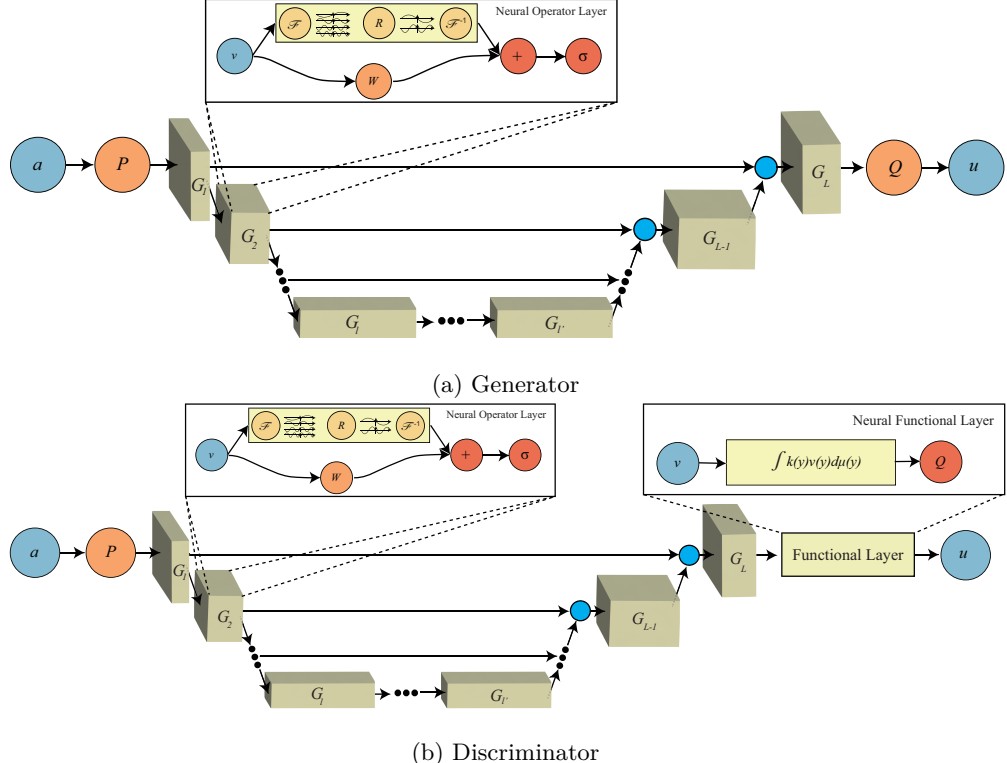

(a) Generator

(b) Discriminator

Figure 1: GANO

Note that, while the cost functional in Eq. 2 is well defined, showing that the learned measure is indeed an approximation of $\mathbb{P}_{\mathcal{U}}$ remains an open problem. We address this issue empirically and perform a set of experiments that demonstrate that GANO produces diverse outputs from the data probability measure.

## 3.2 GANO Architecture

Neural Operators are deep learning models that are the building blocks of the generator and discriminator architectures in GANO to learn maps between function spaces, and the space of reals. In this work, we utilize the *U-NO* architecture (Rahman et al., 2022) for its efficiency, stability, and robustness to the choice of hyper parameters. We implement the generator operator $\mathcal{G}$ using a five layered neural operator model. The inputs to the $\mathcal{G}$ model are samples generated from a GRF defined on the 2D domain of $[0, 1]$. The output of $\mathcal{G}$ are sample functions that defined on 2D domain.

The discriminator is a neural functional that consists of a five layer neural operator followed by an integral functional that maps the output function of the neural operator to a number in $\mathbb{R}$. In other words, we feed an input function $u \in \mathcal{U}$ to the neural operator part of the discriminator to compute the intermediate function $h$ and the output of the discriminator is computed as

$$d(u) = \int k_d(x)h(x)dx \qquad (3)$$

where $d(u) \in \mathbb{R}$, and the function $k_d$ is parameterized as a 3-layered fully connected neural network. The function $k_d$ constitutes the integral functional $\int k_d(x)$ which acts point-wise on its input function. Fig. 1 demonstrate the architecture of the generator $\mathcal{G}$ and the discriminator $d$.

We represent the input function on a grid of $m_1 \times m_2$. It allows us to use autograd to compute the gradient penalty for the Wasserstein loss. Following the function space definitions, the gradient penalty using the

autograd call of $\nabla d(u)$ is implemented as $\mathbb{E}_{\mathbb{P}'_{\mathcal{A}}}(||\nabla d(u)|| - 1/(\sqrt{m_1 m_2}))^2$ which is different than the finite dimensional view in GAN.

## 4   Experiments

In this section, we study the performance of GANO when the data is generated from a GRF. We compare the performance of GANO against GAN in this setting. To implement the GAN baseline model, we deploy convolutional neural networks and use Wasserstein loss for the training.

We then study the effect of the roughness and smoothness of the input GRF on the quality of learning probability measures on function spaces. Lastly, we study the performance of GANO on a real-world remote sensing dataset of an active volcano. This is a challenging dataset with often times very low signal-to-noise ratio.

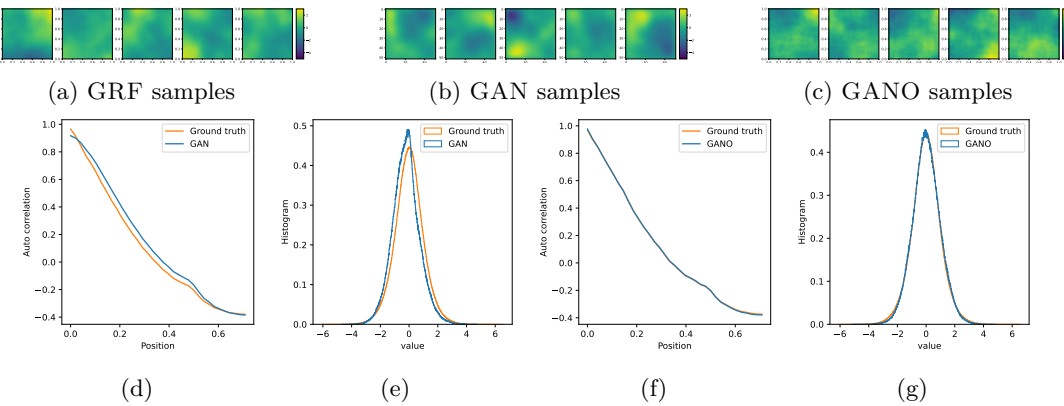

Figure 2: The input function sample is GRF and the data is generated from another GRF. (a) The samples of data GRF. (b) The samples of generated data from GAN model. (c) The samples of generated data from GANO model. (d) GAN Auto correlation. (e) GAN histogram. (f) GANO auto correlation. (g) GANO histogram

**GRF data**. For the setting where data is generated by sampling from a GRF, we use a dataset of random functions drawn from a GRF. We use GAN and GANO approaches to learn the data GRF. We train the generative models using the inputs sampled from a GRF Fig. 2. Fig. 2a demonstrate the sample data. Subsequently, Figs. 2b and 2c demonstrate the generated samples of GAN and GANO models respectively. To analyze the quality of the generated functions, we compare the auto-correlation and histogram of point-wise function values of the generated data and the true data, Fig 2. We observe that GANO properly recovers the statistics of the data GRF in terms of auto-correlation, Fig. 2d, 2f, and the histogram of the generated function values, Figs. 2e, and 2g. We observe that, while the GAN approach provides smoother-looking functions, the functional statistics fail to be exact.

**Mixture of GRFs data**. For this experiment, we aim to learn to generate data from a mixture of GRFs. The training data is generated with an equal chance from either a GRF with a fixed mean function of 1 or $-1$. We use GAN and GANO approaches to learn the data probability measure, where the input functions are sampled from a GRF, Fig. 3. Fig. 2a demonstrate the sample data. Subsequently, Figs. 2b and 2c demonstrate the generated samples of GAN and GANO models respectively. The auto-correlation and histogram of point-wise function values of generated data and the true data are provided in Figs, 2d, 2f 2e, and 2g. As we observe, GANO properly recovers the statistics of the data GRF in terms of functional statistics of auto-correlation and histogram. Similar to the previous experiment, we observe that the GAN approach provides smoother-looking functions, but in terms of the functional statistics, it drastically underperforms GANO.

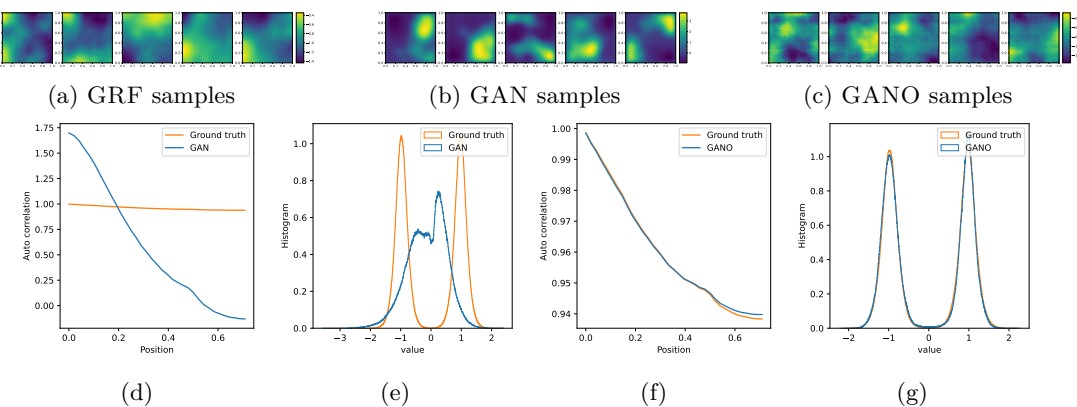

(a) GRF samples    (b) GAN samples    (c) GANO samples

(d)    (e)    (f)    (g)

Figure 3: The input function sample is GRF and the data is generated from a mixture of GRFs. (a) The samples of data from a mixture of GRFs. (b) The samples of generated data from GAN model. (c) The samples of generated data from GANO model. (d) GAN Auto correlation. (e) GAN histogram. (f) GANO Auto correlation. (g) GANO histogram

In the previous two experiments, we observed that GANO enables us to learn measures on function spaces and generate samples that match the functional statistics of the underlying data. In the following, we examine the importance of the choice of input GRF on the performance of GANO.

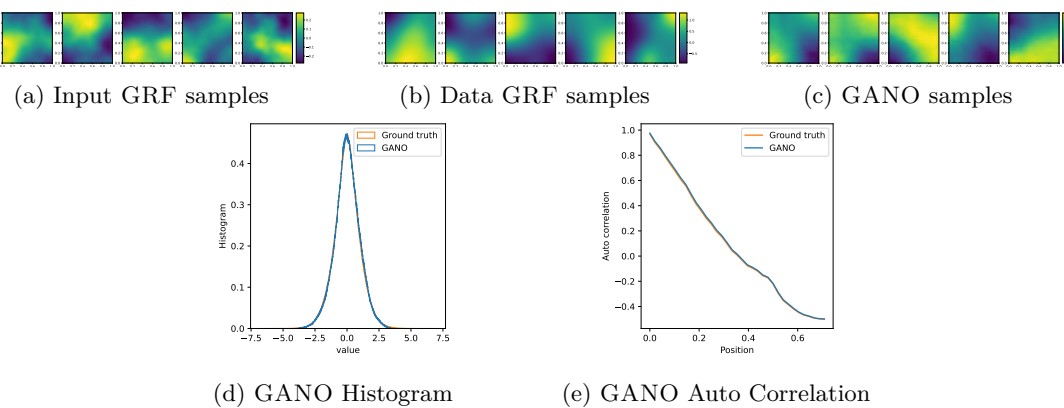

(a) Input GRF samples    (b) Data GRF samples    (c) GANO samples

(d) GANO Histogram    (e) GANO Auto Correlation

Figure 4: GANO trained on smooth data with rougher input GRF

**GANO and the length scale of the input GRF**. In GANO, when the GRF input to the generative model is very smooth (compared with the output GRF), we expect the generator to fail to learn a proper map. We expect this to be the case because the input lacks sufficient high-frequency components, and this smoothness prevents the generator from generating high-frequency and rough output functions. On the contrary, we expect that when the input GRF is much rougher than the data GRF and contains many high-frequency components, the generator would have an easier task to generate output functions. Therefore, the length scale and smoothness of the input GRF can play a role in regularizing GANO model, a very similar role that the dimension of the input multivariate Gaussian plays in the GAN approach.

We first show that when the input GRF is rougher and contains more high frequency components than the output GRF, GANO successfully learns to generate samples with similar statistic of data GRF, Fig. 4.

When the output and input GRF are identical measures, GANO still successfully learns to generate samples with similar statistics of the data GRF, Fig. 5. However, this setting requires more delicate hyper parameter tuning and requires more training epochs to converge. It is worth noting that, with proper choices of the spaces, an identity map may also be a solution.

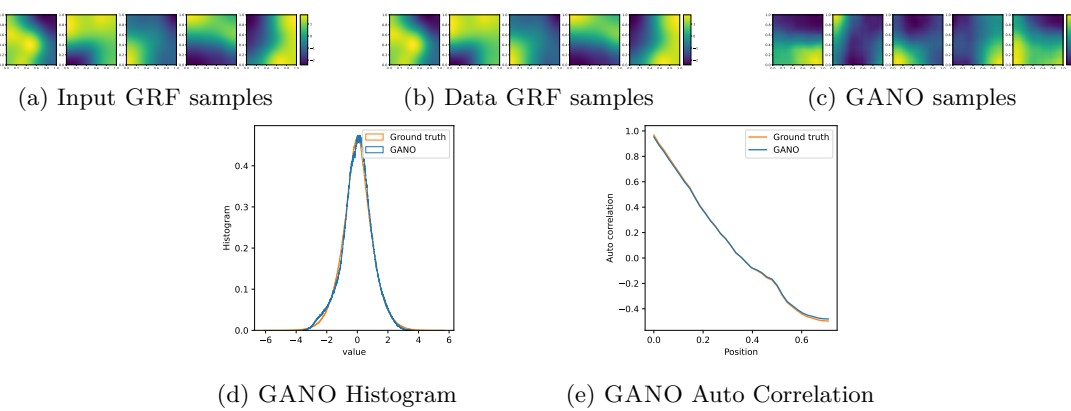

(a) Input GRF samples      (b) Data GRF samples      (c) GANO samples

(d) GANO Histogram      (e) GANO Auto Correlation

Figure 5: GANO trained on same GRF as input and data

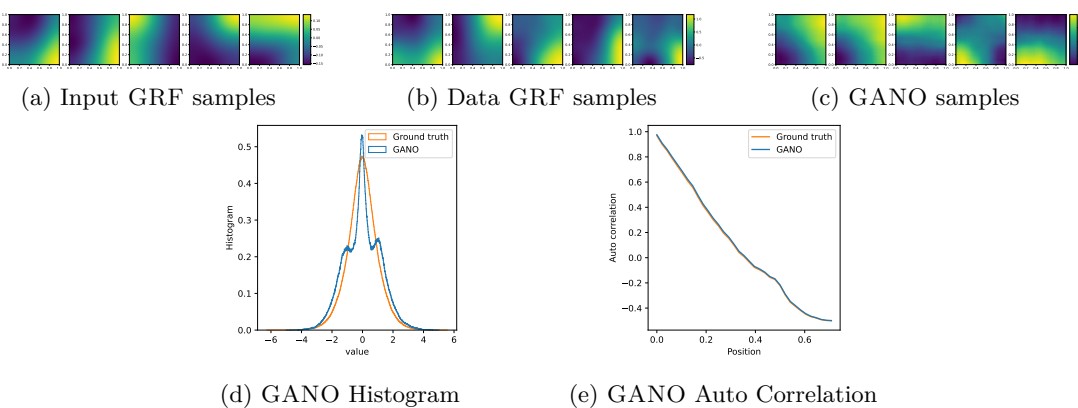

(a) Input GRF samples      (b) Data GRF samples      (c) GANO samples

(d) GANO Histogram      (e) GANO Auto Correlation

Figure 6: GANO trained on rougher data with smoother input GRF

Lastly, when the input GRF is smoother than the functions samples in the output data GRF, the generative model fails to recover higher order statistics, including the auto correlation. In this experiment the input function is much simpler than the output functions. This study suggest that, when the real function data is very complex, very noisy, contains varying high frequency components, and poses high entropy, it is crucial to provide the generator with on par GRF. On the contrary, when the function data at hand poses smoother behavior, a smooth GRF suffices for training a generator.

**Volcano deformation signals in InSAR data.** Interferometric Synthetic Aperture Radar (InSAR) is a remote sensing technology used to measure deformation of Earth's surface, often in response to volcanic eruptions, earthquakes, or subsidence due to excessive groundwater extraction. In InSAR, a radar signal is emitted from satellites or various types of aircraft and echoes are recorded. Changes in these echoes over time (as measured by repeat flyovers) can be used to precisely measure the amount that a point on the surface moves between repeats. The most common form of InSAR data is the interferogram, which is an angular-valued spatial field $u \in \mathcal{U}$, with $u(x) \in [-\pi, \pi]$ and $x \in \mathcal{D}$. Interferograms are known to be highly-complex functions because they exhibit many modalities, types of noises, and patterns that depend strongly on local atmospheric and topographic conditions. Additionally, since the values are angles on $[-\pi, \pi]$, if the change between two echoes is large enough, the angles can wrap around.

We produce a dataset of 4096 data points from raw interferograms, each in a grid of $128 \times 128$, from the Sentinel-1 satellites covering the Long Valley Caldera, which is an active volcano near Mammoth Lakes, California. We processed the InSAR functions/images, publicly provided by the European Space Agency, from 2014-Nov to 2022-Mar, covering an area around Long Valley Caldera (approximately 250 by 160 km

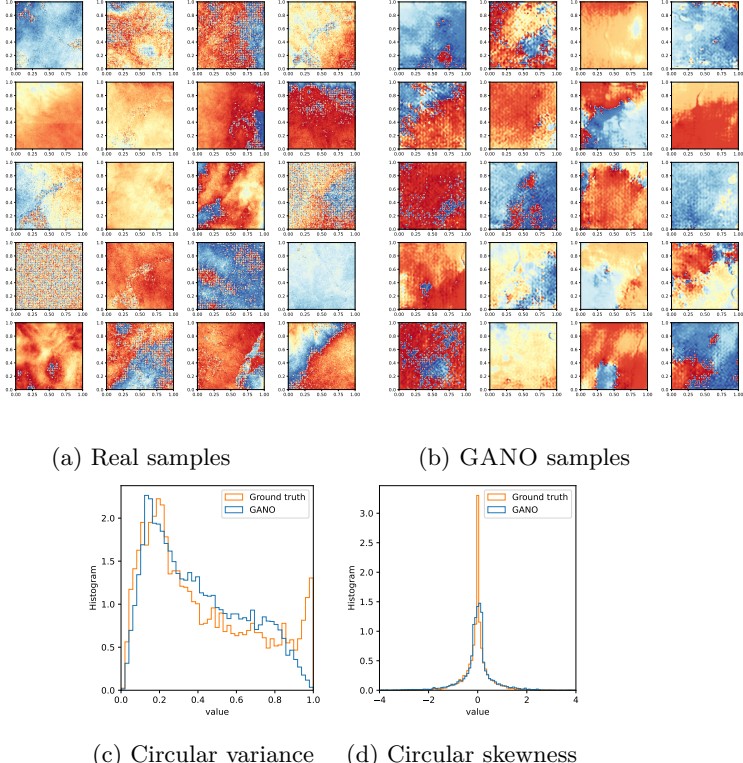

(a) Real samples          (b) GANO samples

(c) Circular variance    (d) Circular skewness

Figure 7: GANO samples of InSAR data for Long Valley Caldera

wide) using the InSAR Scientific Computing Environment (Rosen et al., 2012). The stack of SAR functions is co-registered with pure geometry (precise orbits and digital elevation model) and the network-based enhanced spectral diversity approach. Then, we pair each function (277 in total) with its three nearest neighbors in time to form 783 initial interferograms with pixel spacing of 70 m. Finally, we subset each interferogram into six windows non-overlapping windows of $128 \times 128$ grid. Examples of real samples are shown in Fig. 7a.

We train GANO on the entire dataset of 4096 inteferograms. Generated samples are shown in Fig. 7b, where it is clear that many of the complexities of this dataset have been learned. One of the types of noise in interferograms results from decorrelation of the radar signal between repeat flyovers, and in the most extreme case, can lead to a stochastic process that is random uniform on $[-\pi, \pi]$ that covers part or all of the image. GANO is able to learn an effective operator that approximates this complex behavior. We quantitatively evaluate the quality of the learned samples using circular statistics, which is necessary since these functions are angular-valued. Analogously to traditional random variables, there are moments of circular random variables. For a collection of $N$ random angular variables, $\{\theta_i\}_{i=1}^N$, define $z_p = \sum_j^N e^{ip\theta}$, where $i = \sqrt{-1}$. Then, $R_p = |z_p|/N$ and $\varphi_p = \arg(z_p)$. The circular variance is then given by $\sigma = 1 - R_1$, and the circular skewness is given by, $s = \frac{R_2 \sin(\varphi_2 - 2\varphi_1)}{(1-R_1)^{3/2}}$. Figs. 7c and 7d show the performance of GANO w.r.t circular variance and circular skewness. These results demonstrate the suitability of GANO framework on learning complex probabilities on function spaces and emphasizes the data efficiency of this framework.

For the comparison study, we train a GAN model on the same data set. Despite extensive hyperparameter tuning, the GAN model fails to learn to generate proper samples of functions. Fig. 8b demonstrates samples generated using a trained GAN model. The generated samples do not resemble the true samples, neither perceptually nor with respect to the circular variance and skewness Fig. 8c,8d. This study establishes the

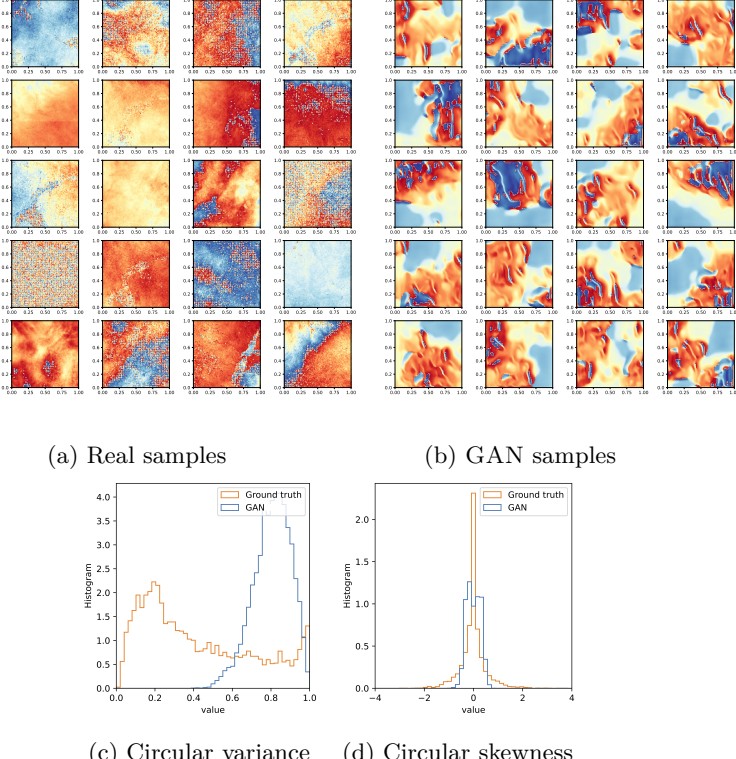

(a) Real samples  (b) GAN samples

(c) Circular variance  (d) Circular skewness

Figure 8: GAN samples of InSAR data for Long Valley Caldera

importance of learning the generative model directly in function spaces using global kernel integration instead of local kernels.

## 5 Conclusion

We propose GANO, a generative adversarial learning approach for learning probabilities on function spaces and generating samples of functions. GANO generalizes GAN, an established and powerful method for learning generative models on finite-dimensional samples. GANO framework consists of two models, a generator operator and a discriminator functional. We use the neural operator framework to directly model the generator and deploy the ideas from neural operators, and propose a new deep learning paradigm, namely neural functional, for the discriminator. We empirically show that the GANO framework is suitable for dealing with function spaces. We show that the input to the generative model can be chosen to be a GRF for which the length scale controls the diversity of the pushed measure. We release the code, package, datasets, and the results of this study for future reproducibility.

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
