# OpenReview forum: "Generative Adversarial Neural Operators"
_TMLR — Rejected by TMLR_

### Review · Reviewer_NvM2 · 2022-06-14

**Summary Of Contributions:**

This submission places the neural operator [1] in an adversarial framework; it proposes training parametrised function space mappings similar to GANs. Empirically, toy examples from Gaussian Random Fields and volcano deformation data (InSAR) are used to demonstrate the capacity of the new model, and its advantages over a GAN baseline.

[1] Li et al. 2018. Neural Operator: Graph Kernel Network for Partial Differential Equations

**Broader Impact Concerns:**

No ethical concern as far as I can tell.

**Requested Changes:**

Addressing the weakness mentioned above is critical to securing my recommendation for acceptance.

**Strengths And Weaknesses:**

### Strength

1. Learning generative models in function spaces is interesting, and the combination of neural operators with GANs looks promising.
2. The volcano deformation data demonstrates the potential practical impact of this work.
3. The paper is overall well-motivated.

### Weakness

1. Important experimental details are missing, which makes comparison between models difficult. For example, are the GANO and GAN baselines comparable in their capacity and computational budget? I notice that source code is provided – this is appreciated, but a summary of the architectural and training details should still be provided in the main text for readability.
2. The paper could be made more self-contained. A formal description of the construction of the generator and discriminator from neural operators would be helpful.
3. While basic GAN serves as the only baseline model, additional baselines especially those operating in function space (e.g, [2]) would better justify the model design.

[2] Kidger et al. 2021. Neural SDEs as Infinite-Dimensional GANs

---

> ### Author Response · Authors · 2022-06-23
> **Learning generative models in function spaces**
>
> Dear NvM2,
>
> We appreciate that the reviewer found learning generative models in function spaces to be interesting and promising to our field. We found the reviewer's comments to be very constructive and they helped in the improvement of the paper.
>
> 1- We incorporate the experiment details into the paper. We provide the architectural details and the number of layers used for both cases. For the comparison, we made the models have the same budget of parameters (equal up to an error of 0.001 in counting the number of parameters). We provide the results in the updated draft. The results indicate the importance of the function space view of GANO.
>
> 2- We provide an explanation of the neural operator, its construction, and layers, along with details of the generator and discriminator to make it self-contained. We also expand on the captions of figure 1.
>
> 3- The paper [2] provides a method that generates 1d-domain signals in time. The resulting generated signals are causal and the discriminator itself is limited to causal functionals where the optimality is left to be cleared. The discriminator loss in [2] is computed with respect to a finite-dimensional space, making the discrimination take place in a finite-dimensional space. In contrast, the GANO framework allows for generating functions in any domain dimension and is not limited to causal data. In GANO, neither the discriminator nor the generator needs to be limited to be causal. More importantly, the discriminator loss is computed in function space rather than finite-dimensional spaces, making the model in fact a generative model for infinite-dimensional function spaces.

---

### Review · Reviewer_e22k · 2022-06-15

**Summary Of Contributions:**

This paper generalizes the well-known generative adversarial network (more precisely Wasserstein GAN) to learn probability distributions over function spaces. Both the generator and discriminator in the proposed GANO framework were built on top of recent advances on neural operator frameworks, where an additional integral layer is introduced in the discriminator for transforming an input function sample to a real number. After introducing the implementation of GANO, the paper then empirically studied the performance of GANO on synthetic datasets generated by GRF and mixture of GRFs, as well as a real function dataset for volcano deformation. The empirical results demonstrate the advantage of GANO over the traditional GAN for function generation in terms of two designed metrics - Histograms and Auto Correlation.

**Broader Impact Concerns:**

No major ethical concerns.

**Requested Changes:**

See the weaknesses section and minor issues section above.

**Strengths And Weaknesses:**

Strengths
- This paper studies an interesting and a relatively new problem - generative modeling for functions, which has the potential for advancing many fields of science and engineering where many data lies in function spaces.
- The empirical study shows clear advantage over traditional GANs.
- Codes and datasets are released for reproducibility.

Weaknesses (and Questions)
- While introducing the architecture, why is the neural functional layer is designed as an integral functional? I understand that we need a way to transform a function into a real number, but any intuition/justification/explanation for such a choice?
Also the RHS of Eq 3 currently has no explicit dependence on the input function sample u on the LHS. also need to justify why we should take this integral form or add this integral layer.
- Can you elaborate more on the gradient penalty calculation in your case (at the end of Sec 3)? If representing the input function on a grid means using a vectors to collect all the m1 x m2 input-output pairs, and then calculate gradient penalty same as finite dimensional vector-valued samples, then this seems to be a straightforward reduction to the usual case and may incur large approximation error since your sample is now infinite-dimensional. Otherwise I am curious about how we should define and calculate gradient penalty on infinite-dimensional function spaces and is there any special consideration needed?
- The experiments show advantage over GAN. But GAN was originally designed for finite-dimensional data like images, so I would not consider it as a strong enough baseline. Why is GAN even applicable in this infinite-dimensional setting (more details needed)? Can you include other baselines that were explicitly designed for generating functions, like the ones in your related work or simple ones like Gaussian Process?
- At the end of Sec 3.1, "Note that, while the cost functional in Eq. 2 is well defined, showing that the learned measure is indeed
an approximation of PU remains an open problem" => Why is that? Does that mean, when we find a pushforward operator G that minimizes equation 1, we still cannot guarantee G P_U is equal to P_A?

Minor issues:
- Do you need paired data to compute auto-correlation? in the case of unconditional generation, where you only have two sets of functions, how do you compute this?
- Do you have/assume closed-form representation for the functions that you want to generate? When this is true, maybe traditional generative models can be used to directly generate the parameters given the structural information.
- Eq 1 is not rigorous. d is not a random variable and should be d(u) where u sim P_U.
- In Eq 2, the extra term implies a different constraint: the gradient should be equal to 1 instead of less than 1. If you use a Lagrangian to directly transform the constraint, it should be different from the one in Eq 2.
- No caption for Figure 1. Also the input to discriminator should be u (not a) and the output should be a real number?

---

> ### Author Response · Authors · 2022-06-23
> **Potentials for advancing many fields of science and engineering**
>
> Dear e22k,
>
> We appreciate the reviewer's assessment that generative models for functions are relevant, and importantly have the potentials for advancing many fields of science and engineering where many data lies in function spaces. We further appreciate the reviewer for the constructive comments.
>
> 1- The linear integral functional in the last layer of the discriminator is the direct generalization of the last layer of discriminators in GAN models, to map a function to a number. In many GAN models, the last layer maps a high-dimensional vector to a number. This step is accomplished by a vector-vector inner product. Such a product, in the continuum, is the function-function inner product, i.e., the act of linear integral functional (this is a similar idea introduced in original neural operators works that replace the vector-matrix product in layers of neural network with their continuum version, i.e. kernel integral operators). This also directly follows the Riesz representation theorem (e.g., Walter Rudin’s real and complex analysis book (1974), Theorem 2.14) stating that, under suitable construction, a linear functional (map from infinite dimension to the space of reals) can be written as a linear integral functional. Thanks to the reviewer's comment, we added an explanation of this choice to the paper.
>
> 2- Before getting to this point, to clarify, m1xm2 is the grid size rather than input-output pairs.
> When a function is represented in a grid of m1xm2, it is indeed represented using m1m2 basis functions. These m1m2 bases functions are unit norm functions in the U metric space. Auto-grad implementation computes the directional derivative of “d” with respect to each grid point value, i.e., u_ij for some i\in[m1] and j\in[m2]. However, in function spaces, we need to consider the basis functions when computing directional derivatives in the direction of basis functions. Intuitively, as the resolution increases, e.g., in the limit, changes of d with respect to u_ij should matter less. This is captured by function spaces derivation in this paper. We further this explanation in the paper.
>
> 3- We elaborate on the use of the GAN framework as the baseline. The prior works propose to use delta sequences, which in the core, basically means that, for n data points, use n delta Dirac functions to represent the density. It is a non-parametric and complete memorization approach, does not learn any structure in the data, and is not very suitable for practical purposes.
>
> 4- As stated in the later comments by the reviewer, equation 2 is the Lagrange relaxation of equation 1. This relaxation is a choice made for practical purposes in most of the Wasserstein GAN studies that we follow in GANO.
>
>
> Minor:
> 1- In order to compute the autocorrelation, we only need samples u\sim P_U. There is no need for pairs.
> 2- We did not make any closed-form assumptions. In fact, the samples from GRF don't have closed forms.
> 3- Updated.
> 4- This is a great point. The relaxation is a soft relaxation that is inspired by the practice proposed in Improved Training of Wasserstein GANs that we missed referencing. We elaborate on this in the main text.
> 5- We add captions and explanations for the figure. We add the architecture explanation and update the figure according to the inputs and outputs.

---

### Review · Reviewer_XtVv · 2022-06-15

**Summary Of Contributions:**

This paper proposes the GANO framework, as a generalization of GANs to functional data. In particular, the generator and discriminator in GANO are neural operators in correspondence to functional inputs. To my knowledge, combining GAN with neural operators is new in literature, even though I don't find any significant modification of the neural operator module.

The second contribution mainly focuses on showcase the performance of the proposed GANO. Through the comparison with GANs, the authors demonstrate the superior performance of GANO. Moreover, the distinction between GAN and GANO is amplified on the real-data example, where GANs almost fail.

Combining these two contributions, the paper provides a self-contained study of GANO for generating functional data, and indicating the viable of GANO.

**Broader Impact Concerns:**

No ethical concerns.

**Requested Changes:**

1. In section 3.1, a brief introduction to Gaussian random field may be provided, as it frequently appears in later sections.

2. The discriminator in Figure 1 does not match its definition: $d \in \mathbf{L}$ is a functional from $\mathcal{U}$ to $\mathbb{R}$, while in the figure, the input is $a$ and output is $u$. Moreover, it is better to add more context in the caption of Figure 1. For example, some explanation of the neural operator layer can be given, in case one is not familiar with it.

3. The last paragraph of Section 3.2 may need some elaboration. Can authors comment on why the altered penalty is implemented in autograd? A more interesting question is the altered penalty term depends on the mesh size of discretization. However, this is not discussed in experiments (see also the following comment for more related questions).

4. In both GANO and GANs, if I am correct, discretization of the input domain is inevitable. The influence of the scale of discretization (i.e., size of grid) is not sufficiently discussed in the paper. On the one hand, one may expect that the performance of GANO relies lightly on the discretization, as it meant to learn the functional directly. On the other hand, one may suspect that the scale of discretization has a nonnegligible impact on the performance of GANs, since it directly determines the input data dimension. I am curious to know should we view the grid size as a tuning parameter to compare the performance of GANs and GANO. For clarity, it is also helpful to provide the discretization information, instead of leaving all the details in the submitted code.

5. Judging from test statistics, e.g., auto correlation and histogram, GANO outperforms GANs. However, as the authors noted, GANs provide smoother-looking generated samples. Can authors further comment on whether this indicates some room of improvement, if not weakness, of GANO? By the way, how are "position" and "value" computed in reported figures?

6. (minor question) In InSAR data experiments, I am wondering why we want to generate new samples mimicking the original functional data distribution. Is it beneficial to some downstream tasks, like predicting earthquakes?

**Strengths And Weaknesses:**

Strengths:

The manuscript is well prepared and easy to follow. Many graphical illustrations are used for explanation and convey direct impressions.

Both synthetic data and real-data experiments are provided to support the success of GANO in applications.

I am fond of the concise idea to integrate neural operator with GANs for allowing functional data generation.

Weaknesses:

I will discuss in requested changes section.

---

> ### Author Response · Authors · 2022-06-23
> **A concise idea for generative models in function spaces**
>
> Dear XtVv,
>
> We appreciate that the reviewer found the idea in the paper to be concise, resulting in functional data generation. We thank the reviewer for the suggestions and the comments that result in a considerable improvement in the readability and clarity of this paper.
>
> 1- We add the definition of GRF in the paper.
> 2- That's a great point. We fix the figure accordingly. We add the explanation of the neural operator, architecture blocks, and the spaces in the caption. We also add a self-contained explanation and deviation of neural operators in the main text.
> 3- For a function represented in a grid size of m1xm2, it is indeed represented using m1m2 basis functions. These m1m2 bases functions are unit norm functions in the U metric space. Auto-grad implementation computes the directional derivative of “d” with respect to each grid point value, i.e., u_ij for some i\in[m1] and j\in[m2]. However, in function spaces, we need to consider the basis functions when computing directional derivatives in the direction of basis functions. Intuitively, as the resolution increases, e.g., in the limit, changes of d with respect to u_ij should matter less. This is captured by function spaces derivation in this paper. Therefore, our penalty in function space is resolution invariant, however, since auto-grad is a finite-dimensional operation, the m1m2 consideration is required. We incorporate this explanation in the paper.
>
> This modification allows training GANO on any resolution, fulfilling the premise of being resolution invariance. This, systematically, allows using multiple resolutions for simultaneous training. This property is directly induced by the GANO framework, and the GAN framework does not allow for it. The penalty in function spaces (eq 2) does not depend on the resolution. We show that, when auto-grad (which is based on finite-dimensional directional derivative) is used, the consideration of m1m2 bases functions is needed. Intuitively, as the resolution increases, changes of d with respect to u_ij should matter less.
>
> 4- We provide the discretization information in the main text, and elaborate on it in the experiment section. Varying resolution in neural operators directly affects the accuracy in the computation of the inner integral operators. The reviewer’s assessment is correct. Reducing the resolution prevents the neural operator from properly capturing P_U.
> The resolution usually come with the sensory and observation devices used in practice and is usually pre-given.
>
> 5- One can further the complexity of the neural networks used in GAN to generate better samples. However, such models may not allow for training on multiple resolutions, and as far as GAN frameworks go, for any resolution, there will be a need to tune the penalty coefficient gamma. None of these limitations prevailed in GANO.
> We add the details on position and value. Value histogram is the histogram of the values the functions take on the grid points and the position is the positional distance between points in the 2D point.
>
>
> 6- The InSAR data are often highly corrupted by complex noises that result from a variety of sources. One potential downstream task would be to de-noise the data to better image magma movement through the subsurface of volcanoes. Learning a generative model for these complex noise sources would therefore improve the signal-to-noise ratio.

---

### Comment · Action_Editors · 2022-06-23
**Reminder: 6 days left in discussion phase**

Dear Authors and Reviewers,

Just a friendly reminder that there are 6 days remaining in the discussion phase, and that reviewers' official recommendations will be submitted once this phase has ended.

We encourage starting a dialogue with reviewers and responding to reviews during this period as necessary. Please keep in mind that the later we receive responses, the less time we have to discuss the recommendations and outcomes.

Thank you,
Your Action Editor

---

### Decision · Action_Editors · 2022-07-15

**Recommendation:** Reject

**Comment:**

The paper proposes a new framework, GANO, that extends the application of GANs to functional data via neural operators. This is a natural and novel idea, and all of the reviewed acknowledged its promise based on the content of the paper, which shows results for several . There were some points of technical clarification required by the authors that were largely addressed by the rebuttal.

Some few important points remain unresolved. First, despite GANO being based on neural operators, both GAN and GANO require discretization to work in practice. GANO may be less sensitive to the specific choice of discretization or be less reliant on it, but discretization seems inevitable. Second, as reviewers note, while GANO outperforms GANs, the choice of discretization for the GANs matched that of GANO rather than tuning the hyper-parameter choice for GANs independently. Third, the comparison to GANs is too weak of a baseline for GANO, leaving the reviewers unconvinced. On this last point, the differences between application domains (i.e. that GANO is more general than the paper by Kidger et al.) were acknowledged by the reviewers, but those differences do not justify the lack of a suitable baseline of comparsion for GANO.

The last point about the weak baseline was the most important factor for the final decisions. I advise the authors not to concentrate too much on a specific baseline (i.e. Kidger et al. in particular), but to look for any baseline that would be more appropriate than GANs since GANs are arguably not designed to solve this problem. Also, some of these potential baselines (and the differences in applicability or assumptions required) should be discussed in more detail in the paper.

Other than the points above, the authors seemed rather positive, so I encourage the authors to resubmit once they have been addressed.